# Cell surface glycan engineering reveals that matriglycan alone can recapitulate dystroglycan binding and function

M. Osman Sheikh [1,9], Chantelle J. Capicciotti [1,8,9], Lin Liu [1], Jeremy Praissman [1], Dahai Ding[1,2], Daniel G. Mead[3], Melinda A. Brindley [3], Tobias Willer[4,5], Kevin P. Campbell [4,5], Kelley W. Moremen [1,6], Lance Wells [1,6 ✉] & Geert-Jan Boons [1,2,7 ✉]

α-Dystroglycan (α-DG) is uniquely modified on *O*-mannose sites by a repeating disaccharide (-Xylα1,3-GlcAβ1,3-)$_n$ termed matriglycan, which is a receptor for laminin-G domain-containing proteins and employed by old-world arenaviruses for infection. Using chemoenzymatically synthesized matriglycans printed as a microarray, we demonstrate length-dependent binding to Laminin, Lassa virus GP1, and the clinically-important antibody IIH6. Utilizing an enzymatic engineering approach, an *N*-linked glycoprotein was converted into a IIH6-positive Laminin-binding glycoprotein. Engineering of the surface of cells deficient for either α-DG or *O*-mannosylation with matriglycans of sufficient length recovers infection with a Lassa-pseudovirus. Finally, free matriglycan in a dose and length dependent manner inhibits viral infection of wildtype cells. These results indicate that matriglycan alone is necessary and sufficient for IIH6 staining, Laminin and LASV GP1 binding, and Lassa-pseudovirus infection and support a model in which it is a tunable receptor for which increasing chain length enhances ligand-binding capacity.

[1] Complex Carbohydrate Research Center, University of Georgia, Athens, GA, USA. [2] Department of Chemistry, University of Georgia, Athens, GA, USA. [3] College of Veterinary Medicine, University of Georgia, Athens, GA, USA. [4] Howard Hughes Medical Institute, Senator Paul D. Wellstone Muscular Dystrophy Specialized Research Center, Department of Molecular Physiology and Biophysics, The University of Iowa, Iowa City, IA, USA. [5] Department of Neurology, Roy J. and Lucille A. Carver College of Medicine, The University of Iowa, Iowa City, IA, USA. [6] Department of Biochemistry & Molecular Biology, University of Georgia, Athens, GA, USA. [7] Department of Chemical Biology and Drug Discovery, Utrecht University, Utrecht, The Netherlands. [8] Present address: Departments of Chemistry, Biomedical and Molecular Sciences, and Surgery, Queen's University, Kingston, ON, Canada. [9] These authors contributed equally: M. Osman Sheikh, Chantelle J. Capicciotti. ✉email: lwells@ccrc.uga.edu; gjboons@ccrc.uga.edu

Dystroglycan (DG) is a highly glycosylated receptor involved in physiological processes such as maintenance of skeletal muscle-cell membrane integrity, signal transduction, brain development, and preservation of neuronal synapses[1,2]. It is post-translationally cleaved into an extracellular α-subunit (α-DG) that is non-covalently linked to a transmembrane β-subunit (β-DG). The intracellular domain of β-DG interacts with several cytosolic proteins, most notably with the structural protein, dystrophin, which in turn binds the actin cytoskeleton[3]. Through their participation in what is known as the dystrophin-glycoprotein complex (DGC), α-DG and β-DG provide a critical glycosylation-dependent link between the extracellular matrix (ECM) and the actin cytoskeleton, especially in muscle tissue[3,4].

Specific O-glycans on α-DG serve as receptors for laminin-G (LG) domain-containing (ECM) proteins such as laminin, agrin, perlecan and neurexin[5,6]. Improper glycosylation of α-DG, due to mutations in genes encoding the involved glycosyltransferases or the enzymes associated with sugar-nucleotide donor biosynthesis, leads to multiple forms of congenital muscular dystrophies collectively referred to as secondary or tertiary dystroglycanopathies, respectively[7–9]. Furthermore, certain arenaviruses, such as Lassa virus (LASV), have evolved cell surface glycoproteins with LG-domains that utilize the same O-glycan structures on α-DG as a receptor to gain entry into host cells[10,11]. LASV causes severe hemorrhagic fever in humans with a mortality rate approaching 15 to 20% in hospitalized patients resulting in thousands of deaths each year in West Africa[12,13]. The virus is carried by rodents of the *Mastomys* genus, and human infection occurs mainly via reservoir-to-human transmission[14]. Due to the high fatality rate, lack of a vaccine, and limited therapeutic options, LASV is considered an important emerging pathogen. Understanding viral entry at a molecular level may provide opportunities for therapeutic intervention.

Although α-DG contains a mucin-like domain rich in O-linked mannosides (Man) and N-acetylgalactosamine (GalNAc) initiating glycans, only three sites (T317, T319, and T379) carry receptors for LG-containing proteins. These sites are modified by a functionally relevant O-Man core M3 glycan that so far has only been observed on α-DG (Fig. 1a)[2,9,15–25]. The O-Man residues destined to be extended and contain laminin-binding sites are extended by POMGNT2 and B3GALNT2 to form the M3 structure (GalNAcβ(1-3)GlcNAc β(1-4)Man-O-Thr), which is then phosphorylated by POMK at the C-6 hydroxyl of the mannoside in the endoplasmic reticulum[20,26–28]. The resulting phospho-trisaccharide is further modified by the Golgi-resident enzymes fukutin (FKTN) and fukutin-related protein (FKRP) to install two phosphodiester-linked ribitol units[22,23]. Next, the enzymes RXYLT1 (formerly known as TMEM5) and B4GAT1 add a xylose (Xyl) and glucuronic acid (GlcA) residue, respectively, to the terminal ribitol-5-phosphate resulting in a GlcAβ(1-4)Xyl priming moiety at the non-reducing end of the M3 glycan structure[9,21,24,25]. This priming structure is a substrate for the bifunctional glycosyltransferase LARGE1, or its paralog LARGE2, which has both xylosyltransferase and glucuronyltransferase activity, and assembles an oligomer composed of [-3Xylα(1,3) GlcAβ1-] repeating units[17,18,29,30]. The latter structure is referred to as matriglycan and serves as the receptor for LG domain-containing ECM proteins, the clinically-relevant monoclonal antibody IIH6, and old-world arenaviruses such as lymphocytic choriomeningitis virus (LCMV) and Lassa virus (LASV)[1,3,31,32]. Importantly, cells defective in any of the post-ribitol glycosyltransferases have reduced molecular weight, no longer bind IIH6, and have a complete loss of laminin binding (Fig. 1b).

The length of matriglycan varies in a developmental and tissue-specific manner, as shown by marked differences in α-DG apparent molecular weight on reducing SDS-PAGE[2]. During myogenesis, the molecular weight of α-DG and the expression levels of DAG1 and LARGE1 increase at the same time[33]. In both cultured cells and mice, ectopic expression of LARGE1 leads to substantial increases in the level of glycosylation of α-DG, which in turn increases the potential to bind ligands of the ECM[33–35]. Alternatively, the non-reducing end GlcA of matriglycan can be capped by sulfation by the enzyme HNK-1ST, preventing further extension[36–38]. In brain, matriglycan has the smallest number of repeating units and the highest ratio of expression of HNK-1ST to LARGE1[38]. Collectively, these findings support a model in which the expression of LARGE1 and HNK-1ST controls the length of

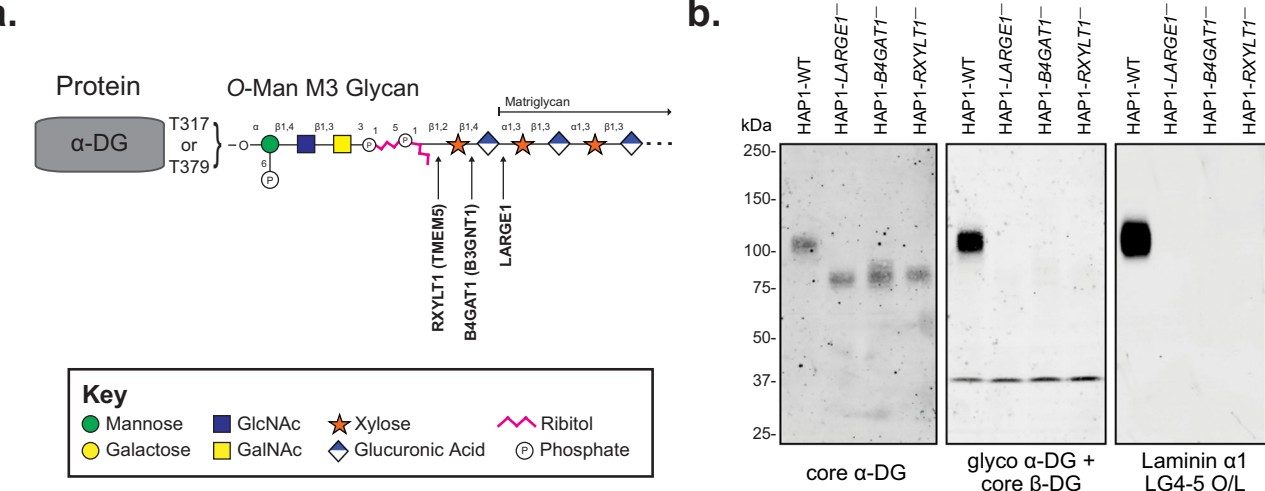

**Fig. 1 Functional Glycosylation of α-Dystroglycan. a** Cartoon representation of the fully elaborated O-mannose M3 glycan that is present on 2 sites of α-dystroglycan with the three post-ribitol enzymes needed for priming and synthesis of matriglycan shown. Carbohydrate symbol representation is consistent with *Symbol Nomenclature for Graphical Representations of Glycans*[71]. **b** Endogenous α-DG was examined in human HAP1 cells (HAP1-WT) as well as HAP1 cells with genetic defects in the post-ribitol enzymes RXYLT1, B4GAT1, or LARGE1. The molecular weight of α-DG was greatly diminished in the cells lacking the post-ribitol glycosyltransferases and was no longer reactive with the IIH6 antibody. α-DG from the three cell lines lacking these enzymes also displayed a complete loss of binding to laminin in an overlay assay. These results highlight the importance of post-ribitol glycosyltransferases for functional glycosylation of α-DG. Three independent experiments were performed with similar results each time.

matriglycan, which in turn, regulates the binding of LG domain-containing proteins.

Despite these observations, it has not been established how many repeating units are needed to bind LG domain-containing proteins. It is also not known whether the protein component of α-DG or the underlying M3 glycan are required for all its functions. One study demonstrated, however, that high molecular weight synthesized LARGE-glycan chains, but not low, are capable of binding laminin-111 and the antibody IIH6, while another more recent study found that a pentamer based on the non-reducing end of matriglycan is capable of binding to laminin-a2 LG 4-5[39,40]. Other studies have utilized overexpression of LARGE, the matriglycan polymerase, in various knock-out cell lines to demonstrate that other glycoproteins containing O-GalNAc and/or complex N-linked glycans can become IIH6 reactive[41,42]. These observations indicate that the protein and underlying M3 glycan structure may be dispensable for certain functions and that matriglycan binding is length-dependent.

The importance of glycan structure and the underlying polypeptide for binding and biological activity is difficult to study because glycan biosynthesis is a non-template mediated process, and therefore conventional genetic approaches do not allow modulation of glycan structures in a systematic manner[43–46]. Glycan array technology, in which hundreds of well-defined compounds are printed on a surface, have been instrumental in establishing binding partners of glycan-binding proteins[47,48]. These assays, however, only report on binding, which does not necessarily correlate with biological function. Thus, additional approaches are required to establish structure-function relationships in the context of cellular processes.

Here, we describe a combined approach of chemoenzymatic synthesis, glycan array technology, cell-surface glyco-engineering, and functional assays to examine whether matriglycan is necessary and sufficient to facilitate LG-domain recognition events of α-DG. It has revealed that presentation of matriglycan, in a length-dependent manner as a receptor for LG-domain binding protein, is not contingent on the underlying O-mannose glycan structure nor the α-DG protein.

## Results

**Matriglycan synthesis and glycan microarray screening.** Matriglycan oligosaccharides with a defined number of repeating units were prepared chemoenzymatically to develop a glycan microarray to establish structure-binding relationships for LG-domain binding proteins. First, xyloside **1** (Fig. 2a) was synthesized, which has an anomeric aminopentyl-linker for immobilization of the glycans to a carboxy reactive surface for microarray fabrication. We opted for a strategy in which **1** was primed by the enzyme β-1,4-glucuronyltransferase (B4GAT1) to provide the disaccharide GlcA-β1,4-Xyl (**2**), which is an appropriate substrate for the glycosyltransferase LARGE1[9,21]. It was anticipated that exposing **2** to LARGE1 in the presence of excess UDP-Xyl and UDP-GlcA would result in the formation of oligosaccharides having different numbers of repeating units. Fractionation of the mixture would then give a range of oligosaccharides of different chain length.

Thus, compound **1** was primed by B4GAT1 in the presence of UDP-GlcA to provide **2** in a near quantitative yield, which was exposed to LARGE1 in the presence of UDP-GlcA (21 eq) and UDP-Xyl (20 eq). A slight excess of UDP-GlcA relative to UDP-Xyl was employed to ensure that each structure terminated with the same monosaccharide unit. Analysis of the reaction mixture by electrospray ionization mass spectrometry (ESI-MS) revealed the presence of oligosaccharides having 2–14 repeating units. Matriglycans with 2–8 disaccharide repeating units (Figs. 2a, 3a–g) could

readily be fractionated by semi-preparative HPLC using a Waters XBridge BEH Amide hydrophilic interaction liquid chromatography (HILIC) column and ESI-MS for detection (Supplementary Fig. 1, Supplementary Table 1, Supplementary Information). Separation by HILIC-HPLC was more challenging for compounds with nine or more repeating disaccharide units and these matriglycans were isolated as mixtures of 9–11 (**3h**) and 12–14 (**3i**) repeating units. Each compound terminated in a glucuronic acid moiety which may be due to the slight excess of UDP-GlcA, but possibly also due to the higher catalytic activity for GlcA transfer at the reaction used conditions, most notably the pH (MES buffered solution, pH 6.0)[18].

Matriglycans **3a-i** were printed on N-hydroxysuccinimide (NHS) activated glass slides. The resulting slides were exposed to different concentrations of the anti-α-DG antibody IIH6, the α-DG binding protein laminin LG4/5, and LASV glycoprotein 1 (LASV GP1). The antibody IIH6 is widely employed to detect functional glycans on α-DG, however, its ligand requirements regarding matriglycan length have not been established. Co-crystallization and NMR binding studies have demonstrated that a matriglycan pentasaccharide can bind to the laminin globular (LG) 4-domain, but laminin has not been examined in microarray binding studies with defined matriglycans[40]. No- or weak binding to the antibody IIH6 was observed for matriglycans with less than 4 repeating units (**2** and **3a-b**, 2–8 monosaccharide units). Interestingly, a compound with 4 repeating units (**3c**, 10 monosaccharide units) was well recognized by the antibody IIH6 and binding gradually increased as the matriglycans were elongated (**3d-i**), despite the absence of the α-DG polypeptide or the underlying O-Man M3 core (Fig. 2b). Both laminin LG4/5 and GP1 LASV GP1 also displayed length-dependent binding (Fig. 2c, d) to the printed matriglycans. No binding was observed for secondary antibody alone.

**Cell-surface glyco-engineering with well-defined matriglycans.** The microarray studies indicated that the binding of various proteins to matriglycan is length-dependent requiring at least ~4 repeating units (n = 4, decasaccharide). To establish whether the structure-binding data correlates with biological function, we sought to modify the plasma membrane extracellular surface of human HAP1-DAG1⁻ cells with well-defined matriglycans for functional studies. These cells have a mutation in the DAG1 gene, which encodes α-DG[11], and therefore do not present matriglycan on α-DG at the cell membrane surface[49]. We opted for a cell-surface glycan engineering strategy that utilizes recombinant ST6GAL1 and CMP-Neu5Ac derivatives modified at C-5 with a bi-functional entity composed of a matriglycan of defined length and biotin. The approach exploits the finding that ST6GAL1 tolerates modification at C-5 of CMP-Neu5Ac and can readily transfer a modified sialic acid to N-linked glycoprotein acceptors of living cells[50]. The biotin moiety provides a handle for monitoring the cell-surface labeling with concurrent structure-function analysis.

CMP-Neu5Ac derivatives **10a-i** (Fig. 3a) were prepared by a convergent strategy where the bifunctional entities **8a-i** were assembled first, followed by conjugation to C5-azide functionalized CMP-Neu5NAz (**9**) by copper-catalyzed alkyne−azide cycloaddition (CuAAC)[51]. The late-stage conjugation made it possible to preserve the labile sugar-nucleotide donor. Thus, xylose derivative **1** was reacted with NHS-activated propargyl glycine (**4**) in the presence of DIPEA to install an alkyne functionality (**1a**) and the resulting compound was immediately treated with Et₃N to remove the Fmoc protecting group (**1b**). The amine of **1b** was reacted with an NHS-activated biotin **5** to give bifunctional **6** having a biotin and alkyne moiety. The primer

**a.**

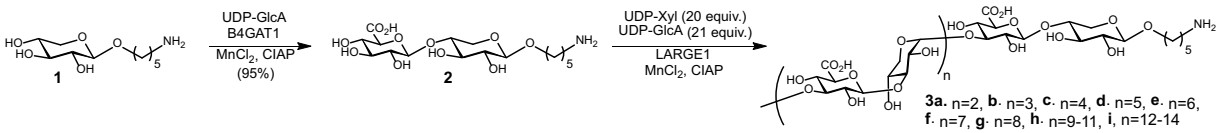

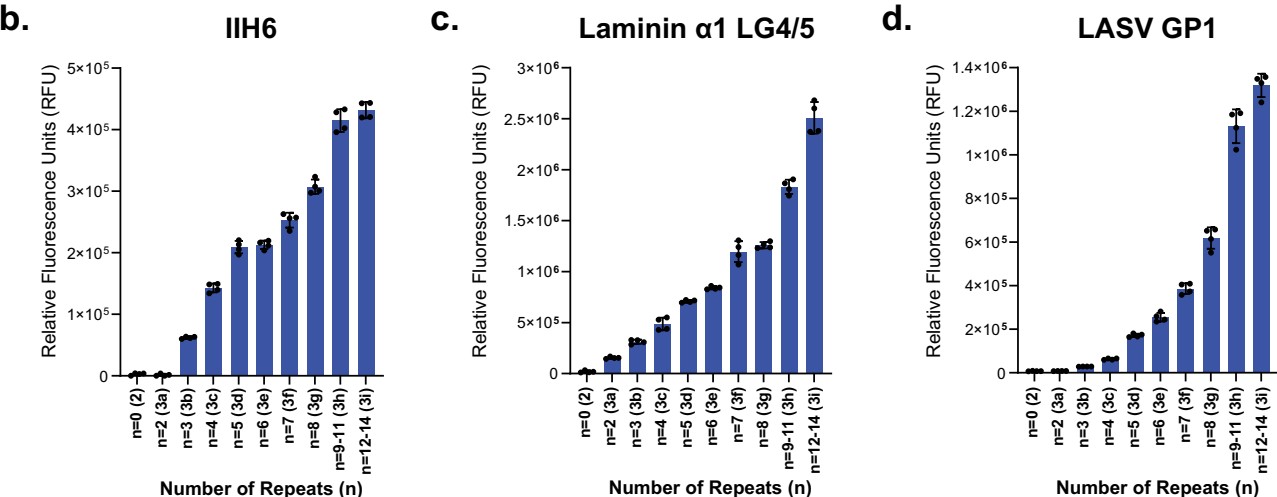

**Fig. 2 Matriglycan microarrays of defined lengths for binding studies. a** Chemoenzymatic synthesis scheme of defined lengths of matriglycan.
**b–d** Microarray binding results with the matriglycan library at 100 μM utilizing **b** mAb IIH6 at 5 μg mL$^{-1}$, **c** recombinantly-tagged LG4-LG5 domains of mouse Laminin α1 (His$_8$-GFP-Lama1) at 20 μg mL$^{-1}$, and **d** Recombinant LASV GP1 protein at 100 μg mL$^{-1}$. For **b–d**, measurements were taken at $n = 4$ technical replicates, where bars represent the mean and error bars represent SD. Source data of **b–d** are provided as a Source Data file.

disaccharide **7** was then obtained by enzymatic modification of **6** with B4GAT1 and UDP-GlcA. Next, **7** was extended with LARGE1 in the presence of excess UDP-Xyl and UDP-GlcA, and after fractionation by HPLC over a HILIC column, bifunctional matriglycans **8a-i** with 1–9 disaccharide repeating units were obtained (Supplementary Fig. 2, Supplementary Table 2). Each compound was conjugated to CMP-Neu5Az (**9**) by CuAAC in the presence of copper sulfate, sodium ascorbate, and Tris[(1-benzyl-1*H*-1,2,3-triazol-4-yl)methyl]amine (TBTA) to afford defined matriglycan-modified CMP-Neu5Ac's **10a-i**.

First, the labeling approach was examined in vitro by determining if the matriglycan-modified CMP-Neu5Ac derivatives can readily modify the model glycoprotein fetuin, which has three *N*-glycosylation sites harboring bi- and tri-antennary complex *N*-glycans terminating in sialosides (Fig. 3b–d, Supplementary Fig. 3)[52], and an *N*-linked glycopeptide isolated from egg yolk powder (Supplementary Figs. 5–7)[53,54]. Thus, fetuin was incubated with ST6GAL1 and CMP-Neu5Ac derivatives **10d-e** (10 eq) in the presence of *C. perfringens* neuraminidase to remove terminal sialosides and create additional *N*-acetyllactosamine (LacNAc) acceptors for ST6GAL1. It is known that a C5-modified sialoside is resistant to neuraminidase cleavage[50,55], and thus neuraminidase and the glyco-engineering step could be performed simultaneously. Western blot analysis of the matriglycan-engineered fetuin's using the anti-α-DG antibody IIH6 indicated that the antibody can recognize matriglycan in the absence of *O*-Man M3 or α-DG-dependent presentation (Fig. 3b). The matriglycan-engineered fetuin's with 4 and 5 repeating units (**10d** and **10e**) were also capable of binding recombinant mouse laminin α1 (LG4-LG5 domains) in an overlay assay (Fig. 3c).

Importantly, neither untreated or neuraminidase-treated fetuin cross-reacted with the IIH6 antibody nor was bound by laminin in the overlay experiment (Fig. 3b, c). Further experiments demonstrated that fetuin remodelled with matriglycans having 1 or 2 repeating units exhibits minimal reactivity in a Western Blot with the IIH6 antibody, while a mixture of 2–10 repeating units showed a strong signal (Supplementary Fig. 3). The strong signal observed with longer matriglycans was abolished by treatment with peptide N-glycosidase F (PNGase F), demonstrating that only N-glycans were modified (Supplementary Fig. 3). The latter is in agreement with previous studies that have shown that ST6GAL1 mainly modifies N-linked glycans[56–58].

To further demonstrate that matriglycan modified CMP-Neu5Ac can be transferred to *N*-glycan acceptors, a bis-galactosylated N-linked glycopeptide substrate (**11**, Supplementary Fig. 4) was isolated from egg yolk powder and enzymatically desialylated to provide a suitable substrate for reaction with ST6GAL1[54]. The resulting *N*-glycan was treated with a mixture of CMP-Neu5Ac derivatives **10a-e** in the presence of ST6GAL1 (Supplementary Figs. 4–6), and the product was analyzed by LC-MS using SeQuant ZIC-HILIC Amide column (Supplementary Fig. 7, Supplementary Table 3). We observed the formation of N-linked glycopeptides (compounds **12a-e**, Supplementary Figs. 4, 7, Supplementary Table 3) modified by a sialoside bearing the expected 1–5 matriglycan repeating units. Only mono-sialylated products were observed, which is consistent with the branch selectivity of ST6GAL1, which prefers the β2-LacNAc over the α3-Man branch of N-glycans[59,60].

Next, attention was focused on engineering cells with well-defined lengths of matriglycan. Thus, HAP1-*DAG1⁻* cells were

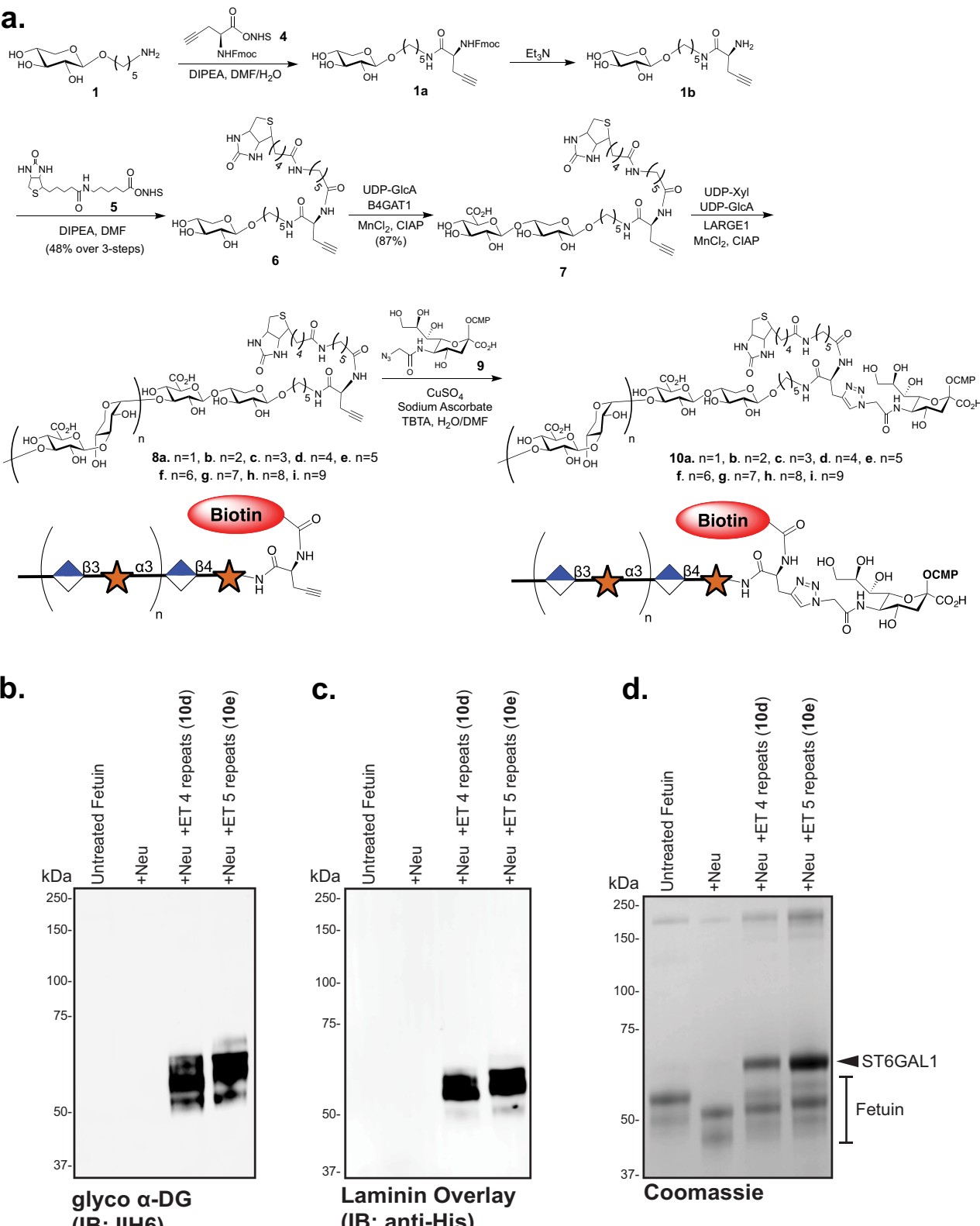

**Fig. 3 Synthesis and testing of CMP-Neu5Ac matriglycan compounds for labeling of N-linked glycoproteins. a** Chemoenzymatic synthesis of bi-functional CMP-Neu5Ac compounds composed of defined matriglycan polymers and a biotin functionality for protein and cell-surface glyco-engineering. Analyses of the enzymatic transfers (ET) of the synthesized compound show positive labeling of fetuin via reactivity with the anti-glyco-α-DG antibody IIH6 **b**, introduces the ability to bind recombinant GFP-His-Laminin LG4/5 as determined by an overlay assay **c**, and retards migration in SDS-PAGE as demonstrated by total protein Coomassie staining **d**. +Neu +Neuraminidase, +ET Enzymatic Transfer by ST6GAL1. Three independent experiments were performed with similar results each time.

incubated with the matriglycan-modified CMP-Neu5Ac derivatives (**10a-i**, 100 μM) in the presence of ST6GAL1 and *C. perfringens* neuraminidase for 2 h at 37 °C (Fig. 4a). First, we confirmed that the matriglycan oligomers were displayed on the surface of HAP1-*DAG1*⁻ cells by avidin staining followed by flow cytometry analysis. While the shorter oligomers gave somewhat more robust labeling suggesting more efficient transfer at equimolar concentrations (Fig. 4b), the results demonstrate that ST6GAL1 can also efficiently transfer the longer glycans including a compound having 6-disaccharide repeating units (**10 f**; 14 monosaccharide units). Next, we examined whether the level of cell surface labeling can be controlled by varying the concentration of the CMP-Neu5Ac derivatives. Thus, different concentrations (1 to 100 μM) of matriglycan-CMP-Neu5Ac derivative **10d** ($n = 4$; 10 monosaccharide units) was exposed to the HAP1-*DAG1*⁻ cells (Fig. 4c). As anticipated, the level of labeling decreased as the concentration was reduced, but was still detectible at 1 μM.

Having confirmed that the CMP-Neu5Ac derivatives can efficiently install well-defined matriglycans on the surface of HAP1-*DAG1*⁻ cells, binding of the IIH6 antibody was examined (Fig. 4d). Cells were labeled with 25 and 100 μM of the CMP-Neu5Ac derivatives and IIH6 binding was assessed by flow cytometry. Antibody binding was only observed for compounds having 5 or more repeating disaccharide units (**10e**; 12 monosaccharide units) and labeling became more robust when the length of the matriglycan increased (Fig. 4d). Even at 100 μM labelling concentration, IIH6 binding was not observed with matriglycan derivative **10d** (4 repeats; 10 monosaccharide units), whereas similar IIH6 binding was observed with **10e** (5 repeats; 12 monosaccharide units) at 25 and 100 μM (Fig. 4d). Cells modified with CMP-Neu5Ac derivative **10i** having 9 repeating units (20 monosaccharide units) bound IIH6 only slightly weaker compared to wild type HAP1 cells that express endogenous α-DG (Fig. 4d)[2]. While there are likely more proteins harboring complex *N*-glycans than there are α-DG matriglycan sites, the chain length of matriglycan on α-DG may be longer thereby providing additional binding sites. The data was employed to choose lengths, concentrations, and labeling conditions for subsequent experiments that do not provide for more IIH6 reactive sites in the HAP1-*DAG1*⁻ cells than present on wild-type HAP1 cells (Fig. 4d).

We enriched glycoproteins on HAP1-*DAG1*⁻ cells that were labeled with matriglycan **10 h** having 8 repeating units by immunoprecipitation using an anti-biotin antibody and performed LC-MS/MS proteomic analysis. For comparison, HAP1-*DAG1*⁻ cells were labeled in a similar fashion with biotinylated CMP-Neu5Ac that does not present a matriglycan moiety. Proteins were identified at a 1% false-discovery rate, and those identified in the negative controls were excluded from the final protein list. A similar subset of *N*-linked glycoproteins was labeled by ST6GAL1 using both modified CMP-Neu5Ac donors indicating that specificity of the enzyme is not substantially altered by using different modified donors (Fig. 4e). The spectral counts for CMP-Neu5Ac modified with biotin alone was higher which agrees with the finding that more complex derivatives transfer less efficiently.

**Cell-surface glyco-engineering rescues LASV infection**. Next, we sought to uncover the minimum number of repeating units required for matriglycan to elicit function. Towards this end, we employed a LASV-pseudovirus entry assay using a recombinant pseudotyped vesicular stomatitis virus (rVSV) in which the glycoprotein (GP) is replaced with that of LASV (rVSV-ΔG-LASV)[11]. The rVSV-ΔG-LASV contains a gene sequence for an

enhanced green fluorescent protein (eGFP) that is utilized as a reporter and for quantification of infection. Using this assay, HAP1 wild type (WT) cells are readily infected by rVSV-ΔG-LASV in an α-DG-dependent manner, whereas HAP1-*DAG1*⁻ cells resist infection. The infectivity of rVSV-ΔG-LASV (MOI 1) was assessed by fluorescence microscopy and quantifying the number of GFP-positive cells 24 h post infection using a Nexcelom Cellometer.

Matriglycans comprised of 2–9 disaccharide repeating units (**10a-i**) were displayed on HAP1-*DAG1*⁻ cells at concentrations ranging from 0.1 to 100 μM using ST6GAL1 in the presence of *C. perfringens* neuraminidase. Remarkably, the cell surface glycan engineering could restore infectivity in a length- and concentration-dependent manner (Fig. 5a, b). At the lowest labeling concentration (0.1 μM), only the longest matriglycan assessed (**10i**, n = 9; 20 monosaccharides) restored infectivity, which was 80% of WT cells. At higher labeling concentrations, additional compounds rescued infectivity. At the highest labeling concentration assessed (100 μM), an oligosaccharide having 4 repeating units could rescue infectivity but not shorter ones. Although shorter matriglycans are more efficiently transferred (Fig. 4), they have reduced or no activity, highlighting the importance of matriglycan length for infectivity. To further validate these findings and to demonstrate that *O*-mannosylation is not substantially involved in the labeling process, we infected HAP1-*POMT2*⁻ cells that are deficient in classical *O*-mannosylation. Infection was blocked in the HAP1-*POMT2*⁻ cells compared to WT, but partially restored by labeling of the *N*-linked glycans with a matriglycan with 6 repeating disaccharide units (**10 f**; 14 monosaccharide units; Fig. 5c).

**Defined soluble matriglycans can inhibit LASV infection in wildtype cells**. Previous studies have demonstrated that soluble, purified α-DG can inhibit LASV infection in vitro[31]. To determine if the synthetic matriglycans can act as a decoy receptor, we employed the matriglycans from Fig. 2a (**3a-e**) as inhibitors of α-DG-mediated viral infection of WT HAP1 cells. Thus, the cells were exposed to rVSV-ΔG-LASV (MOI 1) in the presence or absence of matriglycans having 0, 2, 4, and 6 repeating disaccharides at various concentrations (Fig. 6). Matriglycans with 0 or 2 repeats displayed no to little inhibition of infectivity. Matriglycans having 4 and 6 repeating units were potent inhibitors with IC₅₀ values of 11.2 ± 0.7 and 3.2 ± 0.5 μM, respectively. Thus, substantial inhibition of infection was achieved with free matriglycans in the absence of the extended M3 glycan or the α-DG polypeptide.

## Discussion
Although most mammalian cells express α-DG core protein, its functional glycosylation is under strict tissue-specific control. There is data to support that the matriglycan component of α-DG is a tunable scaffold for LG domain-containing proteins, and by controlling matriglycan length, cells may regulate the recruitment and strength of interaction with such extracellular matrix proteins[39]. Furthermore, there are indications that the α-DG protein is not required for all of matriglycan's functions[61].

Here, we demonstrate that matriglycan is both necessary and sufficient for binding to the clinically-useful IIH6 antibody and LG domain-containing proteins (Figs. 1 and 2). Furthermore, the binding of matriglycan to IIH6, laminin LG4/5, and LASV GP1 is length-dependent, requiring at least 4–5 repeating disaccharide units (10–12 monosaccharide residues, Fig. 2). In each case, the binding gradually increased when the oligosaccharide became longer. Importantly, similar structure-binding profiles were observed for the matriglycans presented on a microarray surface

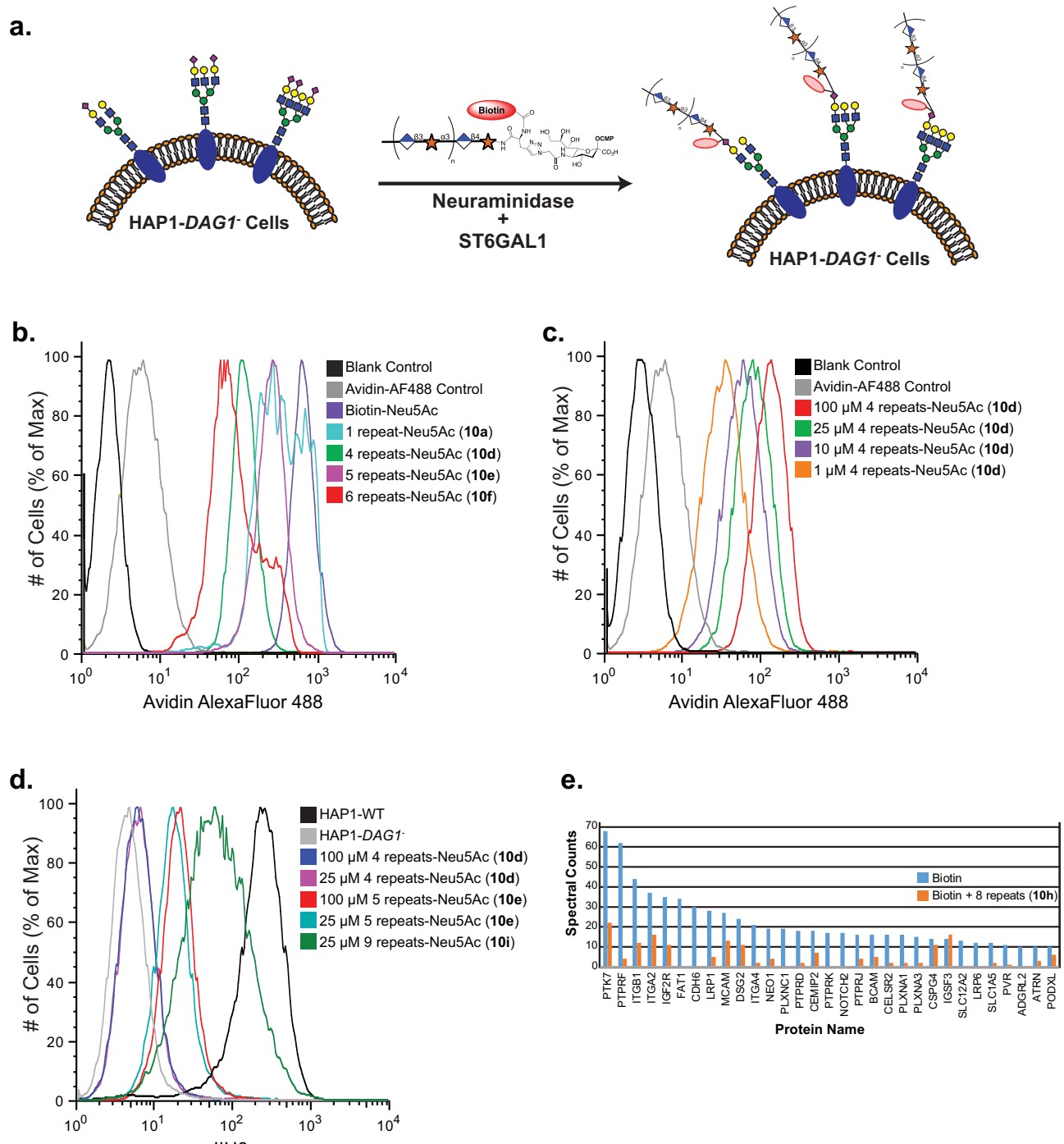

**Fig. 4 Detection of matriglycan on HAP1-*DAG1* cells by flow cytometry. a** CMP-Neu5Ac's modified with defined matriglycan polymeric repeats (100 μM) are engineered on HAP1-*DAG1* cells using ST6GAL1 in the presence of *C. perfringens* neuraminidase. **b** Detection of matriglycan with 1, 4, 5, and 6 disaccharide repeats on HAP1-*DAG1* cells by flow cytometry. Cells were stained with avidin-AF488 and co-stained with PI to exclude non-viable cells. **c** Detection of matriglycan with 4 repeats at various concentrations of modified donor. **d** Binding of IIH6 to HAP1-WT and HAP1-*DAG1* matriglycan modified cells. **e** Shotgun proteomics analysis of proteins immunoprecipitated from HAP1-*DAG1* cells labeled with Biotin or Biotin+8 disaccharide repeats. Proteins present in the negative control experiment (unlabelled cells), had fewer than 10 spectral counts in the CMP-Neu5Ac-(Biotin) labeling experiment, or known to be localized in intracellular compartments as assessed by *UNIPROT* annotations, were excluded. Proteins shown are all annotated in *UNIPROT* to contain sites of *N*-glycosylation or were manually validated to contain at least one N-X-(S/T) *N*-glycosylation sequon in the primary sequence. One representative run is shown. Three independent experiments were performed with similar results each time. Source data are provided as a Source Data file.

(Fig. 2), the N-linked glycoprotein fetuin (Fig. 3, Supplementary Fig. 3), or on a cell surface (Figs. 4 and 5), despite an apparent lower transfer by ST6GAL1 with increasing length at equimolar concentrations. It demonstrates that neither the underlying α-DG

core protein nor the elaborated underlying *O*-mannosylation glycan is required.

It is well-known that anti-carbohydrate antibodies recognize relatively small epitopes ranging from 3 to 7 monosaccharides,

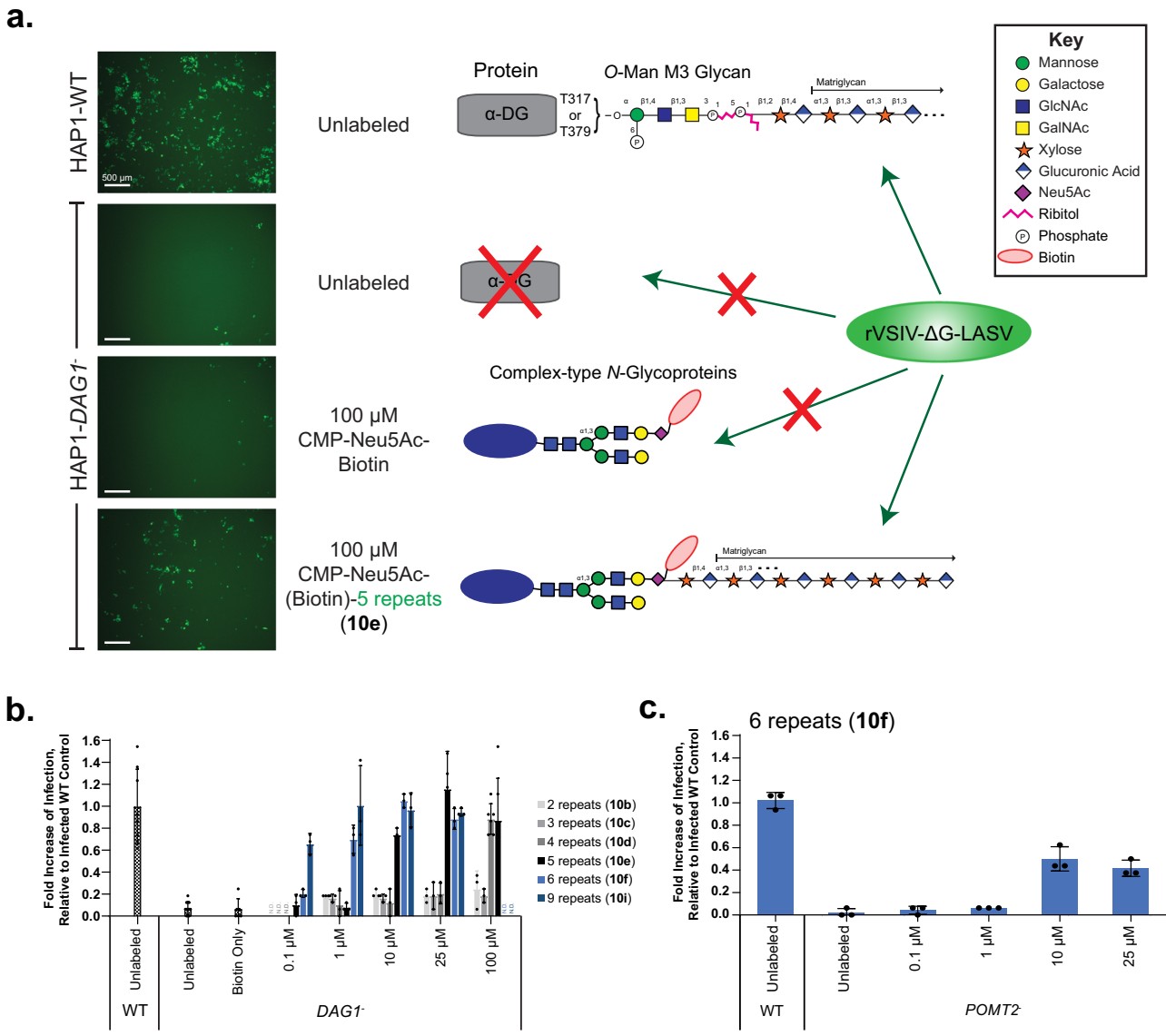

**Fig. 5 Cell Surface Glycan Engineering of matriglycan rescues LASV-pseudovirus infection in dystroglycan-deficient cells.** Matriglycan engineered HAP1-*DAG1*⁻ cells, and unlabeled *DAG1*⁻ and WT controls were incubated for 1 h with rVSIV-ΔG-LASV (MOI 1), then washed and incubated for 24 h before quantifying infectivity. **a** Fluorescence microscopy of eGFP-positive cells 24 h post infection. Three independent experiments were performed with similar results each time. Scale bars represent 500 μm. Also shown are cartoon representations of the glycan structures presented at the surface of indicated cells. **b** Quantification of GFP-positive cells using a Nexcelom Cellometer. **c** LASV-pseudovirus infection of cells lacking the classical *O*-mannosylation pathway (deficient in POMT2), can also be restored by matriglycan labeling with 6 disaccharide repeats. For **b**, **c** measurements were taken at *n* = 3 independent experiments, where bars represent the mean and error bars represent SD. Source data of **b**, **c** are provided as a Source Data file.

and thus it was surprising that IIH6 has a preference for much larger structures. A few examples have been described in which longer oligosaccharides are required for antibody binding. For example, the unusual antigenic properties of meningococcal serogroup B capsular polysaccharide, which is composed of α2,8-linked *N*-acetylneuraminic acid (Neu5Ac) residues, has been ascribed to a conformational epitope requiring at least a decasaccharide to adopt a local helical structure[62]. Previous co-crystallization and NMR binding studies have shown that a matriglycan pentasaccharide is sufficient for binding to the laminin globular (LG) 4-domain[40]. The binding data reported here demonstrate, however, that longer glycans are required for efficient binding by recombinant mouse laminin α1 LG4/5, and binding became more robust with increasing chain length. It is possible that the conformational properties of matriglycan are length-dependent and that a threshold length is required to adopt

a recognition conformation. This possibility is supported by a recent cryo-EM structure of the LASV Spike protein that indicate it can accommodate a matriglycan of at least 13 monosaccharides in a defined spiral-like conformation[63]. It is also conceivable that longer matriglycans can provide a scaffold for multivalent interactions with LG domain-containing proteins resulting in high avidity binding[64]. Previous studies have indicated that in general, at least two sequential LG domains are required for high-affinity binding. Such an assembly of domains has been observed for the LG4/5 of laminins α1, α2, α4, and α5, agrin and pikachurin[64]. Moderate to high-affinity binding by a three LG-domain containing elements has been observed for laminin-α2 and α4 and perlecan. Furthermore, the LASV Spike protein is a trimer with 3 available sites for binding matriglycan[63]. The tandem LG domain found in laminin-α2 of the skeletal muscle isoform and perlecan expressed at the neuromuscular junction, exhibit the highest

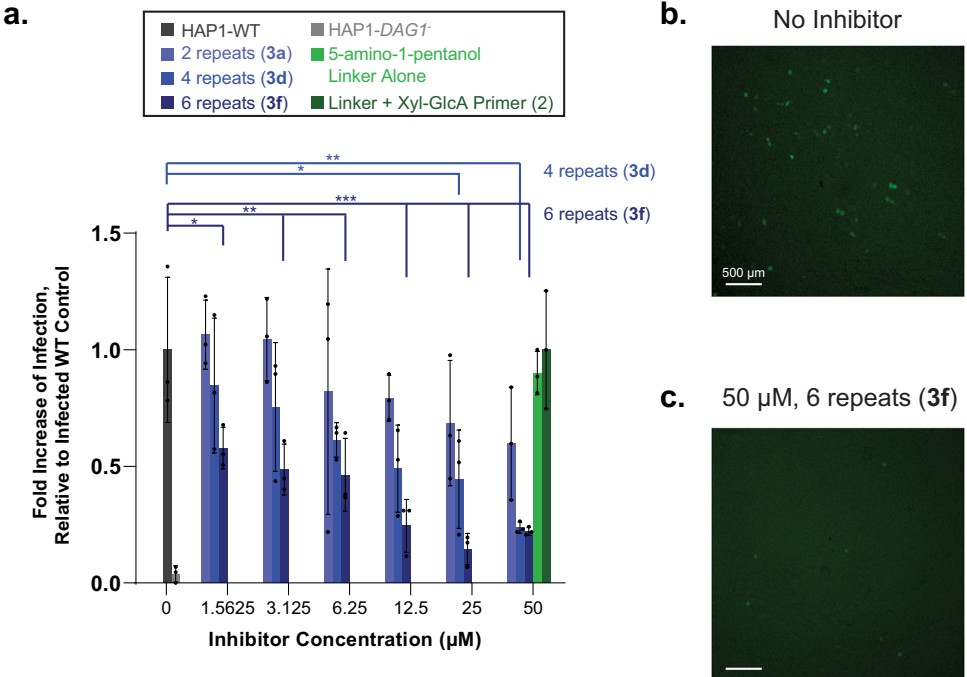

**Fig. 6 Exogenous matriglycan inhibits LASV-pseudovirus infection of wildtype cells in a length and dose dependent manner. a** HAP1-WT cells were infected with rVSV-ΔG-LASV (MOI 1) in the absence and presence of the indicated inhibitor compounds at various concentrations. Infectivity was assessed by fluorescence microscopy by the number of eGFP-positive cells 6 h post infection. The 5-amino-1-pentanol linker compound and the linker compound conjugated to the Xyl-GlcA primer region (0 repeats) were unable to inhibit infection at the highest concentration tested. Measurements were taken at $n = 3$ independent experiments, where bars represent the mean and error bars represent SD. $P$-values are indicated as follows: $P > 0.05$, *$P \leq 0.05$, **$P \leq 0.01$, ***$P \leq 0.001$ (One-way ANOVA followed by Dunnett's Test). **b, c** Representative fluorescence microscopy images of infected WT cells with **b** no inhibitor or **c** 50 μM inhibitor with 6 repeats. Scale bars represent 500 μm. Compound identifiers are defined in Fig. 2 and indicated in parentheses. Source data of **a** are provided as a Source Data file.

affinity binding to dystroglycan, which is in agreement with the biological importance of the DG adhesion complex to stabilize skeletal muscle and the post-synaptic element of the peripheral nervous system. Thus, the presence of tandem arrays of LG domains and the length of matriglycan may provide a way to modulate binding avidity and biological activity.

We also have demonstrated that glycan function can be decoded independently from glycoprotein identity in a cell-based environment as LASV-pseudovirus infection was rescued in α-DG-deficient cells modified with defined matriglycans (Fig. 5). LASV entry is mediated by a glycoprotein complex that is composed of a trimer of heterodimers, each containing a receptor-binding subunit GP1, a transmembrane fusion-mediating subunit GP2 and a signal peptide that has several functions and is retained in the virion as part of the complex. GP1 binds to matriglycan on α-DG to enter the endocytic pathway, where it binds to lysosome-associated membrane protein 1 (LAMP1) before membrane fusion[65]. It is not known whether the protein component of α-DG plays a role in infection but the recent structure of the LASV spike protein demonstrates that it binds matriglycan in the absence of the underlying scaffold[63]. Our results indicate that matriglycan alone is sufficient for LASV-pseudovirus infection in the absence of the protein α-DG and emphasize the functional importance of post-translational glycosylation independent of protein identity. We have also demonstrated that defined soluble matriglycans can function as decoy receptors for LASV infection that is in agreement with recent data that undefined lengths of matriglycan inhibit infection[63], further emphasizing the importance of the terminal matriglycan independent of the underlying glycan or protein structure. Currently, there are limited treatment options for

LASV, and antiviral drugs capable of limiting viral spread may provide the patient's immune system a window of opportunity to develop a protective response. Targeting viral entry is a particularly promising strategy for therapeutic intervention, and matriglycan may offer such a lead compound.

## Methods

**Materials.** All chemical reagents were purchased from Sigma-Aldrich (unless otherwise noted) and used without further purification. All biological reagents were purchased from ThermoFisher Scientific unless otherwise noted. HILIC-HPLC purification of compounds was performed on a Shimadzu 20AD UFLC LCMS-IT-TOF with a Waters XBridge BEH, Amide column, 5 μm, 10 × 250 mm or a SeQuant® ZIC®-HILIC column, 5 μm, 10 × 250 mm. β-Galactoside α-2,6-sialyl-transferase 1 (ST6GAL1) and peptide-N-glycosidase F (PNGase F) were expressed as GFP-fusion constructs as previously described[66]. Calf intestinal alkaline phosphatase (CIAP) was purchased from Sigma. *Clostridium perfringens* (*C. perfringens*) neuraminidase was purchased from New England BioLabs. UDP-Glucuronic Acid was purchased from Sigma. UDP-Xylose was purchased from Carbosource (University of Georgia). HAP1 cells [Parental control (Catalog # C631), *DAG1⁻* (Catalog # HZGHC000120c013), and *POMT2⁻* (Cat # HZGHC003205c001)] were purchased from Horizon Discovery. Alkyne-matriglycan derivatives **8a-i** were stored as lyophilized solids at −20 °C. After lyophilization, matriglycan modified CMP-Neu5Ac's **10a-i** were used immediately for glycoengineering studies.

**Chemoenzymatic synthesis.** Experimental protocols for compounds **1, 4–6,** and **9,** and characterization data for all compounds are provided in the Supplementary Information.

**General procedure for the installation of β1,4-GlcA using B4GAT1.** Xylose acceptor **1** or **6** (10.6 μmol) and UDP-GlcA (15.9 μmmol) were dissolved at a final xylose-derivative concentration of 10 mM in a MOPS buffered solution (100 mM, pH 7.0) containing MnCl₂ (10 mM). CIAP (1% total volume) and B4GAT1 (43 μg/μmol acceptor) were added, and the reaction mixture was incubated overnight at 37 °C with gentle shaking. Reaction progress was monitored by ESI-MS and if starting material remained after 18 h another portion of B4GAT1 was added until

no starting material could be detected. The reaction mixture was centrifuged using a Nanosep® Omega ultrafiltration device (10 kDa MWCO) to remove enzymes and the filtrate was lyophilized. The residue was purified by HPLC using a SeQuant ZIC-HILIC Amide column (5 µm, 10 × 250 mm) with 1% of the flow diverted to the ESI-MS detector (See Supplementary Information). Following HPLC purification, fractions containing product were pooled and lyophilized to yield disaccharides **2** or **7**.

**General procedure for disaccharide extension into matriglycan polysaccharides using LARGE1.** Disaccharide acceptor **2** or **7** (2.0 µmol, 1 equivalent) was dissolved at a concentration of 10 mM in a MES buffered solution (100 mM, pH 6.0) containing $MnCl_2$ (10 mM). For shorter matriglycan lengths ($n < 4$), 4 equivalents of UDP-Xyl (8.0 µmol) and 5 equivalents of UDP-GlcA (10.0 µmol) were added to the reaction mixture. For longer matriglycan lengths ($n > 3$), 17 equivalents of UDP-Xyl (34.0 µmol) and 18 equivalents of UDP-GlcA (36.0 µmol) were added to the reaction mixture. UDP-GlcA was used in excess to cap all matriglycans with GlcA. CIAP (1% total volume) and LARGE (200 µg/µmol acceptor) were added, and the reaction mixture was incubated overnight at 37 °C with gentle shaking. The reaction mixture was centrifuged using a Nanosep® Omega ultrafiltration device (30 kDa MWCO) to remove enzymes and the filtrate was lyophilized. The residue for reactions yielding matriglycans **8** was purified by HPLC using a SeQuant ZIC-HILIC Amide column (5 µm, 10 × 250 mm) (See Supplementary Information). The residue for reactions yielding matriglycans **3** was purified by HPLC using Waters XBridge BEH, Amide column (5 µm, 10 × 250 mm) (See Supplementary Information). Fractions were collected with a volume of approximately 250 µL (20 s intervals) and products were confirmed by ESI-MS before pooling and lyophilizing.

**General protocol for conjugation of matriglycans to CMP-Neu5Az by CuAAC.** Stock solutions of 0.1 M $CuSO_4$, 0.2 M sodium L-ascorbate and 0.1 M TBTA in 0.1 M $NH_4HCO_3$ were freshly made before each CuAAC reaction. 2 equivalents of $CuSO_4$ per GlcA-carboxylate residue were used for each reaction. Sodium ascorbate and TBTA were adjusted to $CuSO_4$ quantities at a ratio of 1.5:1 for sodium ascorbate/$CuSO_4$ and 0.5:1 for TBTA/$CuSO_4$. $CuSO_4$, sodium ascorbate and TBTA were pre-mixed by vortexing, and were then added to a solution of alkyne-matriglycans **8a-i** (1 equivalent) and CMP-Neu5Az **9**[55] (3 equivalents) in 100 µL 0.1 M $NH_4HCO_3$. The resulting mixture was stirred at room temperature for 2 h to have minimal hydrolysis of the CMP-Neu5Ac-derivative. The mixture was then directly loaded onto a P2-BioGel column kept at 4 °C and the product was purified using 0.1 $NH_4HCO_3$ as eluent, analyzed by ESI-MS and immediately lyophilized and used for glyco-engineering studies.

**Microarray procedure.** All compounds were printed on NHS-activated Nexterion® slides purchased from Schott using a Scienion sciFLEXARRAYER S3 non-contact microarray printer equipped with a Scienion PDC80 nozzle (Scienion Inc.). Individual compounds were dissolved in a sodium phosphate buffer (pH 9.0, 250 mM) at a concentration of 100 µM and were printed in replicates of 6 with a spot volume ~400 pL, at 20 °C and 50% humidity. Each slide contained 24 subarrays (3 × 8). Post printing, slides were incubated in a humidity chamber for 24 h and then blocked for 1 h with a 5 mM ethanolamine in a Tris buffer (pH 9.0, 50 mM). Blocked slides were rinsed with DI water, spun dry, and kept in a desiccator at room temperature for future use. Slides were imaged using a GenePix 4000B microarray scanner (Molecular Devices) at the appropriate excitation wavelength with a resolution of 5 µM. The image was analyzed using GenePix Pro 7 software (version 7.2.29.2, Molecular Devices). The data were analyzed with our home written Excel macro to provide the results. The highest and the lowest value of the total fluorescence intensity of the six replicates spots were removed, and the four values in the middle were used to provide the mean value and standard deviation. Raw data values were analyzed and plotted using GraphPad Prism 9.

IIH6 (anti-glyco-α-dystroglycan antibody) screening: The mouse anti-glyco-α-dystroglycan antibody IIH6 (EMD Millipore 05-593) was diluted in a PBS binding buffer (PBSBB: 10 mM PBS, pH 7.4, containing 0.1% BSA and 0.05% Tween) to a final concentration of 5 µg·mL$^{-1}$. IIH6 screening solution (100 µL) was added to the subarray and was incubated at room temperature, in the dark, for 1 h. The slide was washed consecutively with TSM wash buffer (TSMWB: 20 mM Tris-HCl, 150 mM NaCl, 2 mM $CaCl_2$, 2 mM $MgCl_2$, and 0.05% Tween, pH 7.4), TSM buffer (20 mM Tris-HCl, 50 mM NaCl, 2 mM $CaCl_2$, 2 mM, and $MgCl_2$, pH 7.4), DI water, and spun dry. IIH6 was detected by incubating the slide with anti-mouse-IgM-AlexaFluor633 (Invitrogen A21046; 10 µg·mL$^{-1}$ in PBSBB) at room temperature, in the dark, for 1 h. Following incubation, the slide was washed, dried, and visualized.

Laminin LG4/5 screening: Recombinant mouse Laminin alpha 1 LG4-LG5 domains (His$_8$-GFP-Lama1, final concentration 20 µg/mL) was premixed with a biotinylated mouse-anti-His antibody (Invitrogen MA1-21315-BTIN, final concentration 10 µg/mL) in a TBS binding buffer (TBSBB: 25 mM Tris-HCl, 0.15 M NaCl, pH 7.2 with 0.1% BSA and 0.05% Tween) for 15 min. This laminin screening solution (100 µL) was added to the subarray and was incubated at room temperature, in the dark, for 1 h. After washing and drying (as described for IIH6), the slide was then incubated with Streptavidin-AlexaFluor635 (Invitrogen S32364;

5 µg·mL$^{-1}$ in PBSBB) for 1 h in the dark to detect Laminin LG4/5. Following incubation, the plate was washed, dried, and visualized.

GP1 and LASV screening: GP-1 protein was diluted in a TSM binding buffer (TSMBB: 20 mM Tris-HCl, pH 7.4, 150 mM NaCl, 2 mM $CaCl_2$, and 2 mM $MgCl_2$, 0.05% Tween-20, 1% BSA) to a final concentration of 100 µg mL$^{-1}$. The GP-1 solution (100 µL) was added to the subarray and was incubated at room temperature, in the dark, for 1 h. After washing and drying (as described for IIH6), GP-1 was detected by incubating the slide with 2 µg mL$^{-1}$ of Alexa Fluor 633 goat-anti-mouse (H + L) antibody (Invitrogen A-21050) for 3 h. Following incubation, the plate was washed, dried, and visualized.

**Protein expression.** Recombinant expression of soluble, secreted versions of green fluorescent protein (GFP)- B4GAT1 and LARGE1 were expressed and purified as previously described[21,66]. The laminin globular (LG) domains 4 and 5 of mouse Laminin alpha 1 (Gene symbol *LAMA1*, amino acid residues 2705-3083, UniProt P19137) was expressed as a soluble, secreted fusion protein (amino-terminal signal sequence, 8×His-tag, AviTag, and 'superfolder' GFP followed by a TEV-protease cleavage site, referred to as His$_8$-GFP-Lama1). GFP-B4GAT1, GFP-LARGE1, and His$_8$-GFP-Lama1 were all expressed by transient transfection of HEK293F suspension cultures[66]. Suspension culture FreeStyle HEK293F cells (Thermo Fisher Scientific) were transfected as previously described[59]. Six days post transfection, the cell culture media was subjected to Ni-NTA chromatography (Millipore Sigma, St. Louis, MO). The respective proteins were eluted with 300 mM imidazole and concentrated to ~1 mg mL$^{-1}$ using an Amicon centrifugal concentrator (Millipore Sigma, St. Louis, MO) with a 10 kDa molecular weight cutoff and buffer exchanged into PBS pH 7.2.

The LASV GP1 subunit protein coding sequence (amino acids 1–257) was codon optimized for mammalian expression and cloned into a pcDNA3.1 intron vector as a protein fusion with mouse IgG Fc at the carboxy-terminus. The vector (pcDNAintron-LassaGP1-mFc) includes a cytomegalovirus (CMV) promoter, and a β-globin intron was engineered into the 5' untranslated region (UTR) to increase protein production. Suspension culture FreeStyle HEK293F cells (Thermo Fisher Scientific) were transfected as previously described[59]. Six days post transfection, Lassa-GP1-mFc secreted into the cell culture media was purified in batch format using Pierce Protein G agarose (Cat. No. 20398) according to the manufacturer protocol. One to two column volume glycine elution fractions were collected until A280 readings became negligible. The elution fractions were neutralized, pooled, and concentrated at 4 °C using Millipore Microcon-10kDa centrifugal filter units.

**Fetuin glyco-engineering with matriglycan-CMP-Neu5Ac's.** Fetuin (25 µg) was suspended in 50 µL of culture medium without FBS containing the matriglycan-CMP-Neu5Ac derivative (10 equivalents), 10 µg mL$^{-1}$ ST6GAL1, 50 U mL$^{-1}$ *C. perfringens* neuraminidase, 10 U mL$^{-1}$ CIAP and 0.1% BSA for 2 h at 37 °C. Following incubation, samples were stored at −80 °C until analyzed by Western blotting.

**PNGase F treatment.** Fetuin labeled with 1–5 matriglycan repeats (10 µg) was denatured by adding SDS (final concentration 0.25%) and DTT (final concentration 40 mM) followed by incubation at 95 °C for 10 min. After cooling at room temperature for 5 min, Igepal CA-630 was added (final concentration during incubation 1%) along with 1 M HEPES pH 7.3 (final concentration during incubation 50 mM) and pH was checked. The sample was divided equally into two aliquots, PNGase F (2.5 µg) was added to one aliquot, and an equal volume of DI H2O was added to the other aliquot. The resulting samples were incubated overnight at 37 °C and then assayed by SDS-PAGE, Coomassie staining, and western blot.

**Cell culture.** HAP1 cells were cultured in IMDM supplemented with 10% FBS and 1x penicillin/streptomycin. Cells were maintained in a humid 5% $CO_2$ atmosphere at 37 °C and were passaged using 1X trypsin-EDTA or Non-enzymatic Cell Dissociation Buffer and were passaged approximately every 2–3 days (when cells reached 60–80% confluency).

**Cell-surface glyco-engineering with matriglycan-CMP-Neu5Ac's.** HAP1-*DAG1*$^-$ cells were plated in 12-well plates (200 000 cells/well) or 96-well plates (20,000 cells/well) and were grown to 80% confluency. Cells were washed with culture medium without FBS and incubated in a mixture of 300 µL (for 12-well) or 100 µL (for 96-well) culture medium without FBS containing 100 µM (or indicated concentration) of the matriglycan-CMP-Neu5Ac derivative, 10 µg mL$^{-1}$ ST6GAL1, 50 U mL$^{-1}$ *C. perfringens neuraminidase*, 10 U mL$^{-1}$ CIAP and 0.1% BSA for 2 h at 37 °C[50]. Untreated control experiments were treated in a mixture of culture medium without FBS containing 10 µg mL$^{-1}$ ST6GAL1, 10 U mL$^{-1}$ CIAP and 0.1% BSA for 2 h at 37 °C. Following the 2 h incubation time, the matriglycan-engineered cells were washed with 1% FBS/DPBS then treated as indicated.

**Flow cytometry analysis of matriglycan-engineered cells.** For detection of the biotin handle using avidin, matriglycan-engineered cells were stained with avidin-AlexaFluor-488 (Invitrogen A21370; 2.5 µg mL$^{-1}$) in 1% FBS/DPBS for 20 min at

4 °C in the dark. The cells were washed with DPBS without Ca/Mg, then detached using 150 µL of cell dissociation buffer for 2 min at 37 °C. The cells were suspended in 1% FBS/DPBS, centrifuged gently (500 rpm for 3 min), and resuspended in 500 µL of 1% FBS/DPBS and transferred to polystyrene tubes for flow cytometric analysis (Beckman Coulter HyperCyAn, CTEGD Cytometry Center, University of Georgia). Cell viability was determined by adding PI to cell suspensions 5 min prior to analysis. The live population of cells was gated based on forward and side scatter emission, and exclusion of PI positive cells on the FL3 (613/20 BP filter) emission channel. Avidin-AlexaFluor-488 binding was determined by fluorescence intensity on the FL1(530/30 BP filter) emission channel. Data points were collected in duplicates and are representative of two separate experiments ($n = 4$).

For analysis of IIH6 binding, matriglycan-engineered cells were incubated with the anti-glyco-α-dystroglycan antibody IIH6 (EMD Millipore 05-593; 1:250 dilution) in 1% FBS/DPBS for 30 min at 4 °C. Cells were washed, then incubated with goat anti-mouse IgM conjugated with AlexaFluor-488 (Invitrogen A10667; 1:100) for 30 min at 4 °C in the dark. Cells were washed with DPBS without Ca/Mg and were detached, resuspended and analyzed as described above. Data points were collected in duplicates and are representative of two separate experiments ($n = 4$).

**Immunoblotting and laminin overlay assay.** Following SDS-PAGE, proteins were transferred to PVDF-FL (Millipore), blocked with Odyssey Blocking Buffer (Li-Cor), and probed with various antibodies as follows: The anti-α-DG core[67] primary antibody (Goat 20 AP, 1:100 dilution) was detected by secondary antibody donkey anti-goat IgG IR800CW (1:4000, Li-Cor 926-32214). The anti-glyco α-DG[3] primary antibody IIH6 [1:1000 Dilution (EMD Millipore 05-593)] was detected by secondary antibody goat anti-mouse IgM IR800CW (1:4000, Li-Cor 926-32280). The anti-core β-DG mAb 7D11 (1:1000, Santa Cruz sc-33701) was detected by secondary antibody donkey anti-mouse IgG IR680RD (1:10,000, Li-Cor 926-68072). For chain length comparison on matriglycan-engineered Fetuin (Supplementary Fig. 3) immunoblotting for IIH6 was conducted as described above with the following modifications: Odyssey Blocking Buffer was replaced by 0.5% cold water fish gelatin buffers and goat anti-mouse IgM IR800CW (Li-Cor 926-32280) used at 1:10,000 dilution. For immunoblotting, 8 µg of each sample was loaded. For Coomassie staining, 2 µg of each sample was loaded. For the PNGase F sensitivity assay, 4 µg of each sample was loaded for IIH6 immunoblotting and 1 µg of each sample for Coomassie staining.

Laminin overlay assays were performed as previously described, except recombinant His₈-GFP-Lama1 was used[68]. Briefly, following SDS-PAGE, proteins were transferred to PVDF-FL (Millipore), blocked for 1 h with 5% Nonfat Dry Milk in Laminin Binding Buffer [LBB: 10 mM Triethanolamine (TEOA)-HCl pH 7.6, 140 mM NaCl, 1 mM CaCl₂ and 1 mM MgCl₂], and incubated with 10 µg·mL⁻¹ His₈-GFP-Lama1 and 3% BSA in LBB, overnight at 4 °C on an orbital shaker. The following day, membranes were washed in LBB and His₈-GFP-Lama1 was detected using the anti-His.H8 antibody (1:1000, Millipore Sigma 05-949), followed by the secondary antibody donkey anti-mouse IgG IR680RD (1:2000, Li-Cor 926-68072). All immunoblots were imaged using a Li-Cor Odyssey scanner.

**LC-MS/MS proteomic analysis.** After enzymatic cell-surface display of **10 h** on HAP1-*DAG1⁻* cells in 10 cm dishes ($6.5 \times 10^6$ cells/plate), cells were washed with cold DPBS. Cells were lysed by scraping in RIPA buffer supplemented with protease inhibitor cocktail on ice. Lysates were clarified by centrifugation at 22,000 × g for 10 min and the total protein content of the clear supernatants was assessed using the BCA assay. Lysates were immunoprecipitated using protein G beads (Sigma-Aldrich) coated with unconjugated anti-biotin antibody (Jackson ImmunoResearch Laboratories 200-002-211). Coated protein G beads were prepared by incubating the anti-biotin antibody with protein G beads in immunoprecipitation buffer (RIPA buffer without protease inhibitors) at a 3:2 volume ratio of protein G beads: antibody for 2 h at 4 °C. Cell lysates were precleared by incubating with protein G beads for 2 h at 4 °C. The precleared lysate was collected and then incubated with the antibody-coated protein G beads overnight at 4 °C at 1.0 mg of lysate per 50 µL of coated protein G beads. After overnight incubation, the beads were washed 5 times with RIPA buffer and then eluted with 2× sample loading buffer containing 10 mM dithiothreitol by boiling for 10 min. Eluted proteins were resolved by SDS-PAGE and the resulting gel was silver stained for in-gel trypsin digestion followed by proteomic MS analysis.

Each gel lane was excised into four sections above 50 kDa, followed by in-gel tryptic digestion. Proteins in the destained gel sections were reduced by incubation with 10 mM dithiothreitol (Sigma-Aldrich) at 56 °C for 1 h, alkylated with 55 mM iodoacetamide (Sigma-Aldrich) for 45 min in the dark, and digested with Sequencing Grade Trypsin (Promega) at 37 °C overnight. Tryptic peptides were extracted from the gel sections by incubating with increasing concentrations of acetonitrile (25, 50, and 75%, respectively) in 5% formic acid, dried down by centrifugal evaporation, and resuspended in 4% acetonitrile in Solvent A (0.1% formic acid). The peptides were separated using a Thermo Scientific™ UltiMate™ 3000 Rapid Separation Liquid Chromatography (RSLC) system equipped with a 15 cm Acclaim™ PepMap™ RSLC C18 Column [2 µm particle size, 75 µm ID, heated to 35 °C] using a 180 min linear gradient consisting of 1 – 100% Solvent B (80% acetonitrile, 0.1% formic acid) over 130 min at a flow rate of 200 nL/min. Separated peptides were directly eluted into a nanospray ion source of an Orbitrap Fusion Tribrid mass spectrometer (Thermo Fisher Scientific). The stainless steel emitter spray voltage was set to 2200 V, and the temperature of the ion transfer tube was set to 280 °C. Full MS scans were acquired using Orbitrap detection from m/z 200 to 2000 at 60,000 resolution, and MS2 scans following fragmentation by collision-induced dissociation (38% collision energy) were acquired in the ion trap for the most intense ions in "Top Speed" mode within a 3 second cycle using Thermo Xcalibur Instrument Setup (v3.0, Thermo Fisher Scientific). The raw spectra were searched against the Human (*Homo sapiens*) reference proteome database (UNIPROT, Version: "release-2018_08") using SEQUEST HT (Proteome Discoverer v1.4, Thermo Fisher Scientific) with a Full MS precursor mass tolerance of 20 ppm and MS2 peptide fragment mass tolerance of 0.5 Da. Data are reported based on 2 missed cleavages by trypsin, which cleaves at the C-terminal end of Lys and Arg residues, with a minimum peptide length set to at least 6 residues. The search parameters included Carbamidomethylation (on Cys residues) as a fixed modification, and Oxidation (at Met residues) as a variable modification. Protein identifications were filtered using ProteoIQ (v2.7, Premier Biosoft) at the protein level to generate a 5% false-discovery rate (FDR) for peptide assignments and 1% FDR for protein assignments. Proteins present in the negative control experiment (Unlabeled cells), had fewer than 10 spectral counts in the CMP-Neu5Ac-(Biotin) labeling experiment, or known to be localized in intracellular compartments as assessed by UNIPROT annotations, were excluded. Proteins reported are all annotated in UNIPROT to contain sites of *N*-glycosylation or were manually validated to contain at least one N-X-(S/T) *N*-glycosylation sequon in the primary sequence. The mass spectrometry proteomics data have been deposited to the ProteomeXchange Consortium (http://proteomecentral.proteomexchange.org) via the PRIDE partner repository[69] with the dataset identifier PXD024251.

**Infectivity assays using rVSV-ΔG-LASV.** Recombinant VSV expressing eGFP and the Lassa virus glycoprotein (rVSV-ΔG-LASV) was prepared as previously described[11,70]. Following enzymatic cell-surface display in 96-well plate format, cells were gently washed with 100 µL DPBS three times. For each infection experiment, cells from three wells were harvested using 1X trypsin-EDTA and counted using a Nexcelom Cellometer to determine the average cell number per well to determine multiplicity of infection (MOI) calculations. Cells were infected with rVSV-ΔG-LASV at an MOI of 1 (1 virion per cell) in 50 µL of cell culture media for 1 h at 37 °C. Cells were then gently washed with 100 µL DPBS three times, and 100 µL of complete cell culture media was applied to each well. Expression of eGFP was analyzed 24 h post-infection by fluorescence microscopy using a fluorescence microscope (Nikon Eclipse, TE2000-S) and captured using a Qimaging (Retiga 1300i Fast) camera and Qcapture version 2.90.1 software, followed by harvesting of cells and quantification of the number of eGFP-positive cells relative to the total number of cells using a Nexcelom Cellometer. All experiments were performed at technical triplicate or greater. Raw data values were analyzed and plotted using GraphPad Prism 9.

**rVSV-ΔG-LASV inhibition assay.** HAP1 control cells were seeded in 96-well plates (20,000 cells/well) and were grown to 80% confluency. For each infection experiment, cells from three wells were harvested using 1X trypsin-EDTA and counted using a Nexcelom Cellometer to determine the average cell number per well to determine multiplicity of infection (MOI) calculations. For each inhibitor used (matriglycan at different carbohydrate chain lengths), serial dilutions were prepared in cell culture media and mixed 1:1 with twice the concentration of rVSV-ΔG-LASV required to achieve a final MOI of 1 (in 50 µL) at 37 °C for 10 mins. From this inhibitor: rVSV-ΔG-LASV mixture, 50 µL was applied to the respective wells to allow for infection for 1 h at 37 °C. Cells were then gently washed with 100 µL DPBS three times, and 100 µL of complete cell culture media was applied to each well. Expression of eGFP was analyzed 8 h post-infection by fluorescent microscopy using a fluorescence microscope (Nikon Eclipse, TE2000-S) and captured using a Qimaging (Retiga 1300i Fast) camera and Qcapture version 2.90.1 software, followed by harvesting of cells and quantification of the number of eGFP-positive cells relative to the total number of cells using a Nexcelom K2 Cellometer. All experiments were performed at technical triplicate or greater. Raw data values were analyzed and plotted using GraphPad Prism 9.

**Reporting summary.** Further information on research design is available in the Nature Research Reporting Summary linked to this article.

## Data availability

The data that support the findings of this study are available from the corresponding authors upon request. The mass spectrometry proteomics data have been deposited to the ProteomeXchange Consortium (http://proteomecentral.proteomexchange.org) via the PRIDE partner repository[69] under the accession code PXD024251 (LC-MS/MS proteomic analysis of immunoprecipitates). Source data are provided with this paper.

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

## Acknowledgements

K.P.C. is an investigator of the Howard Hughes Medical Institute and L.W. is a Georgia Research Alliance Distinguished Investigator. This work was supported in part by grants from the National Institutes of Health including R01U01GM120408 and R01HL151617 (to G.J.B.), R01GM111939 (to L.W.), R01GM130915 (to K.W.M. and L.W.), R01AI139238 (to M.A.B.), and a Paul D. Wellstone Muscular Dystrophy Specialized Research Center grant (1U54NS053672 to K.P.C.). We would like to thank all the members of all involved laboratories for helpful discussions and especially Dr. Margreet Wolfert for assisting with editing and formatting of the manuscript and associated documents.

## Author contributions

All authors contributed significantly to the manuscript. M.O.S. and C.J.C. contributed equally as co-first authors and were intimately involved in the design of experiments, the generation of reagents, the execution of experiments, the interpretation of data, and the writing and editing of the manuscript. L.L., J.P., D.D., and T.W. were involved in the generation of reagents and execution of experiments. D.G.M., M.A.B., K.P.C., and K.W.M. were involved in the design of experiments, generation of reagents, and interpretation of data. Fiscal support was provided by D.G.M., M.A.B., K.P.C., K.W.M., L.W., and G.J.B. The co-corresponding authors, L.W. and G.J.B., oversaw the entirety of the project and were involved in developing the overall hypotheses being tested, the design of experiments, the interpretation of data, and the writing and editing of the manuscript and associated documents.

## Competing interests

The authors declare no competing interests.
