## [Peer Review File · Nature Communications]

REVIEWER COMMENTS

Reviewer #1 (Remarks to the Author):

Alpha-dystroglycan (alpha-DG) is a highly-glycosylated surface membrane protein containing a laminin-binding glycan (matriglycan, Xyl-GlcA repeats). Reduction or loss of matriglycan is thought to be implicated in muscular dystrophy, cancer, and arenavirus infection. Previously, the author's group reported that the length of matriglycan is important for ligands-binding (laminin-G, LG), but details are unclear. Also, this group is one of leaders of chemoenzymatic synthesis of glycans. Two groups collaborated and examined whether matriglycan is necessary and sufficient to facilitate the LG-domain binding with microarray and cell-surface chemoenzymatic remodeling. They found length dependency of matriglycan binding to LG-domain, and they found that matriglycan is not necessary on alpha-DG and O-mannose glycan. They also reported that free matriglycan in a dose and length dependent manner inhibited viral infection. Although the work is of interest, several concerns could be found and more convincing data are required.

(1) Major concern is that the authors did not show the quantitative transfer-efficiency data of CMP-Neu5Ac derivatives to N-glycans by ST6GAL1. This review wonders whether or not ST6GAL1 shows length-dependent transfer difference. They should provide such data, for example, transfer data of CMP-Neu5Ac derivatives to fixed bi- and/or tri-antennary complex N-glycans. It is essential to interpret following data of this manuscript correctly.

In related to this, only data of IIH6 and laminin-overlay of 4 and 5 repeats were shown in Figure 3. Data of other number of disaccharide-repeats should be provided.

(2) Another concern is that the authors addressed to length of disaccharide units but did not density of disaccharide units on a cell surface. Generally, many kinds of N-glycans attach on each cell-surface protein. And it seems that the density of N-glycosylated proteins is higher than that of alpha-DG, suggesting many proteins are modified by their chemoenzymatic methods. They should provide clear data the effect of not only length of glycan but also density of glycan-modified core proteins on binding LG domain.

They described simply in page 7 line 10 [Interestingly, - - - (Fig. 4d)]. This sentence is obscure and too descriptive. It is unclear what do the authors want to intend. What is biological significance of relationship between 9-repeats on N-glycoproteins and longer glycan on alpha-DG without quantitative data?

(3) Figure 4b ~ d. The authors provided data of flow cytometry. However, several different colored-lines in 4b and d largely invisible in my copy. It hampered this review. In addition, western-blotting with I1H6 and laminin-overlay will be necessary to strengthen their claims.

(4) Figure 4e. Only spectral counts of biotin+8repeats of IGSF3 among listed proteins showed higher than biotin. Why?

(5) It seems to be different levels rescued by 6 disaccharides repeats at 10 and 25 uM (Figure 5b and c). The rescued level of POMT2- is lower than DAG1-, suggesting that core M3 is important? Another possibility is that N-glycosylated proteins on cell-surface are different between two mutant cells (DAG1- and POMT2-). They should show western-blotting with I1H6 and laminin-overlay of these cells.

Figure 5b. Data of 6 and 9 repeats glycans at 100uM should be shown.

(6) Figure 6a. Although the authors concluded matriglycan having 2 repeating disaccharides did not show any inhibition of virus infectivity, but it seems inhibitory at high concentration. Is it correct?

(7) Previously, several groups already reported that LARGE-induced hyperglycosylation is not restricted on O-mannose glycan and alpha-DG (e.g. *Glycobiology*, 19,971,2009; 22,235,2012), but the authors did not refer them and discuss at all. It is unclear whether or not they consider relationship between the present study and the previous studies.

(8) Although authors described that only two sites (T317 and T379) appear to carry structures that function as receptors for LG-containing proteins (page 2), it has been reported that another site (T319) can also be modified with such a type glycan [*PNAS*. 108, 17426–17431, 2011; *Science* 327, 88–92, 2010; *Sci. Rep.* 3, 1–9, 2013, etc.].

(9) If the authors performed a series of experiments using tunicamycin treated cells (no N-glycans) or GnT-1 KO cells (only mannose-type N-glycans), they should provide data. It may strengthen their story.

Minors

(10) Fig. S2 and Fig. 3S may be Fig. 1S and Fig. 2S, respectively.

(11) Data of 3i are missing in Table S1.

(12) Page 6 line 2. STGAL1 is ST6GAL1.

(13) Fig. 5 and 6. Scale bars should be provided.

(14) Fig. 2 and 3. LARGE may be LARGE1.

(15) Fig. 3 legends. Figure 2 key may be Figure 1 key.

(16) h and hour are mixed. Also, many typos could be found in Methods and SUPP.

(17) Refs, 33 and 36, 41 and 49, are duplicate.

Reviewer #2 (Remarks to the Author):

It was claimed in the current manuscript that matriglycan alone is necessary and sufficient for IIH6 staining, Laminin and LASV GP1 binding, and Lassa-pseudovirus infection and support a model in which it is a tunable receptor for which increasing chain length enhances ligand-binding capacity, based on the results of free matriglycan and cell-surface glyco-engineering with well-defined matriglycans. This report is novel, impressive and will attract a lot of interest to others in the community. I suggest it is suitable for publication in this journal after minor revisions.

1. In the figure 3a, derivative 6 was obtained in 48% yield by three steps. It should be better to illustrate the synthetic procedure for three steps on the equation.

2. It is claimed that a-DG contains a mucin-like domain rich in O-linked mannosides (Man) and N-acetylgalactosamine (GalNAc). The relevant references for chemoenzymatic synthesis for O-glycan could be updated.

3. In the supporting information (page S3, line 6), the filtrate was lyophilized to yield compound 1, not 12.

Reviewer #3 (Remarks to the Author):

This manuscript addresses the role of synthetic soluble or conjugated matriglycan versus matriglycan on a-DG/DAG for binding of Ab IIH6, laminin (4/5LG) or GP1 of LASV. It is an elegant and complete experimental system, with all methods clearly explained and reagents extensively characterized. The

binding assays and the infectivity of LASV-VSV pseudovirus are well explained, and the data are appropriately quantitative and sufficient. Based on the combined data, the authors can conclude, in clear and unambiguous terms, that synthetic matriglycan alone, or expressed on a heterogeneous set of cellular glycoproteins in cells lacking DAG, must contain more than 4 LARGE-generated glycan repeats in matriglycan to be well-recognized by the Ab IIH6, Laminin 4/5LG or LASV GP 1. Pseudovirus binding to HAP1 cells lacking DAG but carrying matriglycan on other glycoproteins is functional in that the cells become infected. This is a comprehensive demonstration that matriglycan does not need either the M3 core glycan nor DAG to be recognized by ligands, or to function in LASV entry.

Minor recommendations.

Suggest qualifying "The O-Man cores destined to become the laminin-binding sites---" to be more explanatory. For example: "The O-Man residues destined to be extended and contain laminin-binding sites----"

A word of explanation as to why matriglycan length was not extended beyond n=12-14 would be helpful since the amount bound for each substrate did not plateau.

Signed: Pamela Stanley

We thank the reviewers for their supportive and constructive comments, which have made it possible to improve the manuscript. Reviewers 2 and 3 noted only minor point to be corrected which we have addressed. Reviewer 1 noted several issues requiring additional experiments. Due to turnover of personal, it has taken a bit longer than hoped to perform these experiments. However, we have now completed experiments which we hope addresses all points noted by Reviewer 1. Furthermore, several sections have been rewritten to provide further clarity. The conclusions of the paper are based on *multiple experimental approaches* that support the notion that matriglycan is a tunable receptor and that the underlying M3 glycan structure is dispensable for certain functions.

Reviewer #1 (Remarks to the Author)

Alpha-dystroglycan (alpha-DG) is a highly-glycosylated surface membrane protein containing a laminin-binding glycan (matriglycan, Xyl-GlcA repeats). Reduction or loss of matriglycan is thought to be implicated in muscular dystrophy, cancer, and arenavirus infection. Previously, the author's group reported that the length of matriglycan is important for ligands-binding (laminin-G, LG), but details are unclear. Also, this group is one of leaders of chemoenzymatic synthesis of glycans. Two groups collaborated and examined whether matriglycan is necessary and sufficient to facilitate the LG-domain binding with microarray and cell-surface chemoenzymatic remodeling. They found length dependency of matriglycan binding to LG-domain, and they found that matriglycan is not necessary on alpha-DG and O-mannose glycan. They also reported that free matriglycan in a dose and length dependent manner inhibited viral infection. Although the work is of interest, several concerns could be found and more convincing data are required.

We thank the reviewer for appreciating the importance of our research.

Remark. Major concern is that the authors did not show the quantitative transfer-efficiency data of CMP-Neu5Ac derivatives to N-glycans by ST6GAL1. This reviewer wonders whether or not ST6GAL1 shows length-dependent transfer difference. They should provide such data, for example, transfer data of CMP-Neu5Ac derivatives to fixed bi- and/or tri-antennary complex N-glycans. It is essential to interpret following data of this manuscript correctly. In related to this, only data of IIIH6 and laminin-overlay of 4 and 5 repeats were shown in Figure 3. Data of other number of disaccharide-repeats should be provided.

Response. We have addressed the concerns of the reviewer by performing additional experiments, and by providing further explanations of data reported in the initial submission.

To demonstrate that CMP-Neu5Ac modified by matriglycans can be transferred to a fixed N-glycan acceptor, a bis-galactosylated N-linked glycopeptide substrate (compound **11**, Scheme S1) was isolated from egg yolk powder and enzymatically desialylated to provide a suitable substrate for ST6GAL1. The resulting N-glycan was treated with a mixture of CMP-Neu5Ac derivatives **10a-e** in the presence of ST6GAL1 (Scheme S1, Figs. S4-5), and the product was analyzed by LC-MS using SeQuant ZIC-HILIC Amide column (Fig. S6, Table S3). As expected, we observed the formation of N-linked glycopeptides modified by a sialoside bearing the 1-5 matriglycan repeating units (Scheme S1, Fig. S6, Table S3).

Further experiments have demonstrated that the addition of matriglycan consisting of 1 or 2 repeating units to the N-glycans of fetuin exhibits minimal reactivity in a Western Blot with the

IIH6 antibody, while a mixture of 2-10 repeats showed a strong signal, consistent with the array data (Fig. S3). The strong signal observed with longer matriglycan oligomers could be abolished by treatment with peptide N-glycosidase F (PNGase F), confirming that only N-glycans were modified (Fig. S3).

The flow cytometry data described in the initial submission addresses the issue regarding length dependence of transfer. The CMP-Neu5Ac derivatives are modified by a matriglycan and a biotin moiety, which makes it possible to monitor cell-surface labeling with concurrent structure-function analysis. The data in Fig. 4b, in which avidin is used as probe, shows that shorter matriglycans are more efficiently transferred than the longer ones. Our approach makes it possible to control the level of matriglycan transfer by varying the concentration of the CMP-Neu5Ac derivative (see Fig. 4c). The enrichment/proteomic data (Fig. 4E) using equal concentration of CMP-Neu5Ac derivative also demonstrate that smaller entities are more efficiently transferred. These findings actually augment our argument that IIH6/laminin binding of matriglycan is length dependent. In this respect, although the larger oligosaccharides are less efficiently transferred, they exhibit better binding or biological activity. We have clarified this important point by several statements. The section dealing with cell surface labeling states: “*Although shorter matriglycans are more efficiently transferred (Fig. 4), they have reduced- or no activity, highlighting the importance of matriglycan length for infectivity*”. The discussion notes “*Importantly, similar structure-binding profiles were observed for the matriglycans presented on a microarray surface (Fig. 2), the N-linked glycoprotein fetuin (Fig. 3, Fig. S3), or on a cell surface (Fig. 4 and 5), despite an apparent lower transfer by ST6GAL1 with increasing length at equimolar concentrations. It demonstrates that neither the underlying α -DG core protein nor the elaborated underlying O-mannosylation glycan is required*”.

Remark. Another concern is that the authors addressed to length of disaccharide units but did not density of disaccharide units on a cell surface. Generally, many kinds of N-glycans attach on each cell-surface protein. And it seems that the density of N-glycosylated proteins is higher than that of alpha-DG, suggesting many proteins are modified by their chemoenzymatic methods. They should provide clear data the effect of not only length of glycan but also density of glycan-modified core proteins on binding LG domain.

Response. The biotin handle makes it possible to monitor the density of matriglycan on the engineered cells. Concentrations of the Neu5Ac derivatives were selected in such a way that it would not result in more IIH6 signal in the HAP1-DAG1⁻ cells than in wildtype HAP1 cells (illustrated in Figure 4D). The infectivity studies (Fig. 5) have been performed at multiple concentrations of labeling reagent.

Remark. They described simply in page 7 line 10 [Interestingly, - - - (Fig. 4d)]. This sentence is obscure and too descriptive. It is unclear what do the authors want to intend. What is biological significance of relationship between 9-repeats on N-glycoproteins and longer glycan on alpha-DG without quantitative data?

Response. This point goes hand in hand with the previous point and it may be our lack of clarity in the initial manuscript that led to both points. The statement was not meant to overinterpret the data. It has been reworded and is aimed at clarifying the rationale of utilizing concentrations of

matriglycan labeling in HAP1-*DAG1*⁻ cells. It is stated: “While there are likely more proteins harboring complex *N*-glycans than there are α -DG matriglycan sites, the chain length of matriglycan on α -DG may be longer thereby providing additional binding sites. The data was employed to choose lengths, concentrations, and labeling conditions for subsequent experiments that do not provide for more IIH6 reactive sites in the HAP1-*DAG1*⁻ cells than present on wild-type HAP1 cells (Fig. 4d).”

The conclusions of the paper are based on different experimental formats, including microarray based binding studies, fetuin remodeling followed by binding assays, cell remodeling combined with binding and functional assays. Collectively all the generated data support the notion that matriglycan is a tunable receptor and that the underlying M3 glycan structure is dispensable for certain functions.

Remark. Figure 4b - d. The authors provided data of flow cytometry. However, several different colored-lines in 4b and d largely invisible in my copy. It hampered this review. In addition, western-blotting with IIH6 and laminin-overlay will be necessary to strengthen their claims.

Response. We apologize that Fig. 4b-4d were unclear and have recreated these figures. The data presented in figure 4E is even more powerful than a Western blot as it shows the actual identity of the modified proteins that is also combined with flow cytometry for IIH6 signal (Fig. 4d). Multiple proteins bind to laminin and thus we restricted our analysis of laminin binding to purified matriglycan repeats in the array format (Fig. 2d) or on a recombinant protein (Fig. 3c) to assess the presence of matriglycan (free or bound to N-linked glycoprotein) alone on laminin binding.

Remark. Figure 4e. Only spectral counts of biotin+8repeats of IGSF3 among listed proteins showed higher than biotin. Why?

Response. This is a semi-quantitative assay that was aimed at confirming that similar N-linked glycoproteins are labeled by both reagents. Given the low number of spectral counts observed for IGSF3, it is likely that the difference is not significant.

Remark. It seems to be different levels rescued by 6 disaccharides repeats at 10 and 25 μ M (Figure 5b and c). The rescued level of POMT2⁻ is lower than DAG1⁻, suggesting that core M3 is important? Another possibility is that N-glycosylated proteins on cell-surface are different between two mutant cells (DAG1⁻ and POMT2⁻). They should show western-blotting with IIH6 and laminin-overlay of these cells. Figure 5b. Data of 6 and 9 repeats glycans at 100 μ M should be shown.

Response. It is likely that other co-receptors are involved in LASV infectivity, which may provide a rationale for the differences between the two cell types. The discussion provides a short section on additional co-receptors involved in LASV infection.

We did not show labeling at 100 μ M of 6 and 9 repeating units to not over-label the cells (see above). Furthermore, the longest matriglycans linked to CMP-Neu5Ac are challenging to generate in sufficient quantities for triplicate analyses. As mentioned above, our conclusions are based on multiple and orthogonal experimental approaches.

Remark. Figure 6a. Although the authors concluded matriglycan having 2 repeating disaccharides did not show any inhibition of virus infectivity, but it seems inhibitory at high concentration. Is it correct?

Response. While there was a trend towards 2 repeating units inhibiting at a very high concentration as noted by the reviewer, it did not reach statistical significance like for other lengths as shown in Figure 6A. Nonetheless, the data supports that the potency of inhibition is depended on the length of the matriglycan and is in agreement with rescue and binding studies presented in the preceding sections.

Remark. Previously, several groups already reported that LARGE-induced hyperglycosylation is not restricted on O-mannose glycan and alpha-DG (e.g. *Glycobiology*, 19,971,2009; 22,235,2012), but the authors did not refer them and discuss at all. It is unclear whether or not they consider relationship between the present study and the previous studies.

Response. We thank the reviewer for pointing out this oversight. We have addressed these intriguing findings in the introduction. It is stated “*Other studies have utilized overexpression of LARGE, the matriglycan polymerase, in various knock-out cell lines to demonstrate that other glycoproteins containing O-GalNAc and/or complex N-linked glycans can become IIH6 reactive.⁴¹⁻⁴² These observations indicate that the protein and underlying M3 glycan structure may be dispensable for certain functions and that matriglycan binding is length-dependent.*”

Remark. Although authors described that only two sites (T317 and T379) appear to carry structures that function as receptors for LG-containing proteins (page 2), it has been reported that another site (T319) can also be modified with such a type glycan [PNAS. 108, 17426–17431, 2011; *Science* 327, 88–92, 2010; *Sci. Rep.* 3, 1–9, 2013, etc.].

Response. We thank the reviewer for pointing out this oversight, which has been corrected.

Remark. If the authors performed a series of experiments using tunicamycin treated cells (no N-glycans) or GnT-1 KO cells (only mannose-type N-glycans), they should provide data. It may strengthen their story. Since GnT-1 KO or tunicamycin treated cells would be expected to provide very few if any complex N-linked glycoproteins for ST6Gal1 to act on,

Response. Along the suggested lines, we have included an experiment showing that the labeling of fetuin to make it IIH6 reactive can be ameliorated by PNGaseF treatment confirming the dependence on N-linkage (see Fig. S3).

We thank the reviewer for pointing out a number of minor typographical errors which have been corrected.

Reviewer #2 (Remarks to the Author)

It was claimed in the current manuscript that matriglycan alone is necessary and sufficient for IIH6 staining, Laminin and LASV GP1 binding, and Lassa-pseudovirus infection and support a model

in which it is a tunable receptor for which increasing chain length enhances ligand-binding capacity, based on the results of free matriglycan and cell-surface glyco-engineering with well-defined matriglycans. This report is novel, impressive and will attract a lot of interest to others in the community. I suggest it is suitable for publication in this journal after minor revisions.

We thank the reviewer for the highly positive assessment of our research.

Remark. In the figure 3a, derivative **6** was obtained in 48% yield by three steps. It should be better to illustrate the synthetic procedure for three steps on the equation.

Response. Purification was performed after the three steps which makes reporting individual yields difficult.

Remark. It is claimed that a-DG contains a mucin-like domain rich in O-linked mannosides (Man) and N-acetylgalactosamine (GalNAc). The relevant references for chemoenzymatic synthesis for O-glycan could be updated.

Response. The paper does not deal with chemoenzymatic synthesis of O-glycans and thus such a reference is less relevant. Reference has been made to the preparation of the N-glycan employed in this study.

Remark. In the supporting information (page S3, line 6), the filtrate was lyophilized to yield compound 1, not 12.

Response. Corrected.

Reviewer #3 (Remarks to the Author)

This manuscript addresses the role of synthetic soluble or conjugated matriglycan versus matriglycan on a-DG/DAG for binding of Ab IIH6, laminin (4/5LG) or GP1 of LASV. It is an elegant and complete experimental system, with all methods clearly explained and reagents extensively characterized. The binding assays and the infectivity of LASV-VSV pseudovirus are well explained, and the data are appropriately quantitative and sufficient. Based on the combined data, the authors can conclude, in clear and unambiguous terms, that synthetic matriglycan alone, or expressed on a heterogeneous set of cellular glycoproteins in cells lacking DAG, must contain more than 4 LARGE-generated glycan repeats in matriglycan to be well-recognized by the Ab IIH6, Laminin 4/5LG or LASV GP 1. Pseudovirus binding to HAP1 cells lacking DAG but carrying matriglycan on other glycoproteins is functional in that the cells become infected. This is a comprehensive demonstration that matriglycan does not need either the M3 core glycan nor DAG to be recognized by ligands, or to function in LASV entry.

We thank the reviewer for the positive comments about convincingly supporting the central points of our paper.

Minor recommendations:

Remark. Suggest qualifying “The O-Man cores destined to become the laminin-binding sites---” to be more explanatory. For example: “The O-Man residues destined to be extended and contain laminin-binding sites----”

Response. The sentence has been reworded as suggested to qualify the statement.

Remark. A word of explanation as to why matriglycan length was not extended beyond n=12-14 would be helpful since the amount bound for each substrate did not plateau.

Response. As mentioned above in response to Reviewer 1, we chose to not use lengths or concentrations on HAP1-*DAG1* cells that exceeded IIH6 binding of wildtype HAP1 cells. Also, the largest oligosaccharides are more challenging to produce in quantities required for certain experiments. These points have been clarified in the revised manuscript.

REVIEWERS' COMMENTS

Reviewer #1 (Remarks to the Author):

Overall, the revised manuscript by Sheikh et al. has improved the quality and clarity by extra experimentation and explanation of the text. I recommend for publication after the authors address the minor comments below.

(Fig. S6 and Table S3)

The authors obtained only mono matriglycan-modified biantennary N-glycopeptide after overnight incubation with ST6GAL1 and CMP-Neu5AC derivatives, and explained the results on the basis of branch preference of ST6GAL1. They did not provide any data of both antennae modified N-glycopeptide. However, in Fig. 4a and Fig. 5a including graphical abstract, all antennae of N-glycan (bi-, tri- and tetra) are modified by matriglycan derivatives. Notably, they incubated cells with ST6GAL1 only for 2 h for cell-surface engineering. It may mislead the reader. The authors need the underlying explanation for all antennae of N-glycan modified by matriglycan.

Figure 3a, Structure of glycan derivative of third row left. ()5 in lower arm may be missing.

Page S3. Structure of Matriglycan 8: Lower arm, ()5 may also be missing.

Page S8. In the revised MS, 12 is not fixed as it is 12. It may be 1 as pointed out.

Several "a" of a-DG in Discussion may be Greek letter (alpha).

REVIEWERS' COMMENTS

Reviewer #1 (Remarks to the Author):

Overall, the revised manuscript by Sheikh et al. has improved the quality and clarity by extra experimentation and explanation of the text. I recommend for publication after the authors address the minor comments below.

Response: Thank you for the positive and critical evaluation.

(Fig. S6 and Table S3)

The authors obtained only mono matriglycan-modified biantennary N-glycopeptide after overnight incubation with ST6GAL1 and CMP-Neu5AC derivatives, and explained the results on the basis of branch preference of ST6GAL1. They did not provide any data of both antennae modified N-glycopeptide. However, in Fig. 4a and Fig. 5a including graphical abstract, all antennae of N-glycan (bi-, tri- and tetra) are modified by matriglycan derivatives. Notably, they incubated cells with ST6GAL1 only for 2 h for cell-surface engineering. It may mislead the reader. The authors need the underlying explanation for all antennae of N-glycan modified by matriglycan.

Response: Figure 4a and Figure 5a have been corrected to accurately reflect the substrate specificity of ST6GAL1. The Graphical Abstract has been removed from the manuscript.

Figure 3a, Structure of glycan derivative of third row left. ()5 in lower arm may be missing. Page S3. Structure of Matriglycan 8: Lower arm, ()5 may also be missing.

Response: Figure 3a and Page S3 (Structure of Matriglycan 8) have been corrected.

Page S8. In the revised MS, 12 is not fixed as it is 12. It may be 1 as pointed out.

Response: This has been corrected.

Several "a" of a-DG in Discussion may be Greek letter (alpha).

Response: This has been corrected.